The P2 nucleic acid binding protein of Sugarcane bacilliform virus is a viral pathogenic factor

http://orcid.org/0000-0001-7365-680X Xu Xiongbiao 1 xiongbiaox@gxu.edu.cn
http://orcid.org/0009-0003-5825-6471 Lou Yinian 1
Liang Kaili 1
Liu Jingying 1
Wang Zhiyuan 1
Chen Baoshan 1
Li Wenlan 2
1 State Key Laboratory for Conservation and Utilization of Subtropical Agro-bioresources, Guangxi key Laboratory of Sugarcane biology, Province and Ministry Co-sponsored Collaborative Innovation Center of Canesugar Industry, College of Agriculture, Guangxi University , Nanning, Guangxi , China
2 College of Life Science and Technology, Guangxi University , Nanning, Guangxi , China
Böttcher Bettina
Electronic publication date: 2024 Feb 20
Publication date: 2024
Volume: 12
Electronic Location ID: e16982
Received 2023 Nov 24; Accepted 2024 Jan 30
Copyright: © 2024 Xu et al.
Copyright year: 2024
Copyright holder: Xu et al.
License: This is an open access article distributed under the terms of the Creative Commons Attribution License, which permits unrestricted use, distribution, reproduction and adaptation in any medium and for any purpose provided that it is properly attributed. For attribution, the original author(s), title, publication source (PeerJ) and either DOI or URL of the article must be cited.
License URL: https://creativecommons.org/licenses/by/4.0/

Keywords: Sugarcane bacilliform virus, Phylogenetic relationship, Pathogenic factor, RNA silencing suppressor, Hypersensitive-like response

Funding: Natural Science Foundation of Guangxi Province No. 2023GXNSFBA026058 Sugarcane Research Foundation of Guangxi University No. 2022GZB011 Innovation Project of Guangxi Graduate Education No. YCSW2022065 This work was supported by the Natural Science Foundation of Guangxi Province (No. 2023GXNSFBA026058), the Sugarcane Research Foundation of Guangxi University (No. 2022GZB011) and the Innovation Project of Guangxi Graduate Education (No. YCSW2022065). The funders had no role in study design, data collection and analysis, decision to publish, or preparation of the manuscript.

==============================
Background

Saccharum spp. is the primary source of sugar and plays a significant role in global renewable bioenergy. Sugarcane bacilliform virus (SCBV) is one of the most important viruses infecting sugarcane, causing severe yield losses and quality degradation. It is of great significance to reveal the pathogenesis of SCBV and resistance breeding. However, little is known about the viral virulence factors or RNA silencing suppressors and the molecular mechanism of pathogenesis.

Methods

To systematically investigate the functions of the unknown protein P2 encoded by SCBV ORF2. Phylogenetic analysis was implemented to infer the evolutionary relationship between the P2 of SCBV and other badnaviruses. The precise subcellular localization of P2 was verified in the transient infiltrated Nicotiana benthamiana epidermal mesophyll cells and protoplasts using the Laser scanning confocal microscope (LSCM). The post-transcriptional gene silencing (PTGS) and transcriptional gene silencing (TGS) RNA silencing suppressor activity of P2 was analyzed, respectively. Furthermore, restriction digestion and RT-qPCR assays were conducted to verify the probable mechanism of P2 on repressing DNA methylation. To explore the pathogenicity of P2, a potato virus X-based viral vector was used to heterologously express SCBV P2 and the consequent H2O2 accumulation was detected by the 3,3′-diaminobenzidine (DAB) staining method.

Results

Phylogenetic analysis shows that SCBV has no obvious sequence similarity and low genetic relatedness to Badnavirus and Tungrovirus representatives. LSCM studies show that P2 is localized in both the cytoplasm and nucleus. Moreover, P2 is shown to be a suppressor of PTGS and TGS, which can not only repress ssRNA-induced gene silencing but also disrupt the host RNA-directed DNA methylation (RdDM) pathway. In addition, P2 can trigger an oxidative burst and cause typical hypersensitive-like response (HLR) necrosis in systemic leaves of N. benthamiana when expressed by PVX. Overall, our results laid a foundation for deciphering the molecular mechanism of SCBV pathogenesis and made progress for resistance breeding.

Introduction

Saccharum spp. is a primary sugar-producing crop and an important industrial renewable bioenergy crop cultivated throughout the world’s tropical and subtropical areas. Sugarcane bacilliform virus (SCBV) is one of the primary viruses infecting sugarcane and causing severe damage. It was first identified in cultivar B34104 in Cuba in 1985 (Geijskes et al., 2002), and later purified from cultivar Mex.57-473 (Lockhart, 1988). SCBV is spontaneously spread by the insect vectors Dysmicoccus boninsis and Saccharicoccus sacchari. It can also be transmitted experimentally by raw viral sap or Agrobacterium-mediated inoculation (Lockhart, Irey & Comstock, 1995), but failed to be transmitted by mechanical friction. In addition, long-distance spread of virus-infected materials is an important means of transmission. SCBV has a relatively broad host range, including Sorghum halepense, Brachiaria sp., Rottboellia exaltata, Panicum maximum, and experimental hosts such as Oryza sativa and Musa sp. (Bouhida, Lockhart & Olszewski, 1993; Lockhart, Irey & Comstock, 1995; Viswanathan, Alexander & Garg, 1996). Usually, the visible symptoms caused by SCBV are mottling, stunted growth, chlorotic streaks, and internode fracture (Viswanathan, Alexander & Garg, 1996). Once infected, the juice extraction, sucrose content, gravity purity and average stalk weight were decreased 1.55%, 1.24%, 2.22% and 0.26 kg, respectively, but reducing sugar was increased by 0.21% in infected plants, resulting in significant yield and quality losses (Li et al., 2010; Ahmad et al., 2019). Occasionally, masked symptoms occur due to temperature, drought, and nutritional conditions changes. Moreover, much more complicated symptoms could be found due to co-infection with other viruses (Lockhart, Irey & Comstock, 1995; Singh et al., 2009; Viswanathan & Premachandran, 1998).

SCBV belongs to the genus Badnavirus (family Caulimoviridae), with bacilliform, non-enveloped virions of 130–150 nm in length and 30 nm in diameter (Bhat, Hohn & Selvarajan, 2016; Bouhida, Lockhart & Olszewski, 1993; Lockhart, 1988), containing a circular, covalent, discontinuous dsDNA genome of approximately 7.5–8.0 kilobases (Kb), which generally encodes three open reading frames (ORFs). The exact roles of ORF1 and ORF2 have not been confirmed (Bouhida, Lockhart & Olszewski, 1993), and ORF3 has been found to encode a large polyprotein, which was truncated into movement protein (MP), coat protein (CP), aspartic protease (AP), reverse transcriptase (RT) and ribonuclease H (RNase H), but the precise cutting sites remain unknown (Geijskes et al., 2002; Sun et al., 2016). Previous studies have suggested that a short fragment between the 3′-end of ORF3 and the 5′-end of ORF1 may act as a strong promoter in both monocot and dicot species (Davies et al., 2014; Gao et al., 2017). Commelina yellow mottle virus (CoYMV) and Cacao swollen shoot virus (CSSV) are the two well studied members of genus Badnavirus, family Caulimoviridae. CoYMV is the type member of badnaviruses and the P2 protein has been shown to be involved in virion particle assembly (Cheng, Lockhart & Olszewski, 1996), and the P2 protein encoded by CSSV can bind to cognate or heterologous DNA as well as ssRNA and the C-terminus appears to be essential for nucleic acid binding (Jacquot et al., 1996). Jacquot, Keller & Yot (1997) have demonstrated that the proline-rich region (99-PPKKGIKRKYPA-110) at the C-terminal of Rice tungro bacilliform virus (RTBV) P2 plays a vital role in the interaction between P2 and nucleic acids. All these results provide the basis for studying the function of the P2 protein of Badnavirus, but little is known about its roles in viral pathogenicity.

In this study, the complete genome of SCBV was deciphered, and the functions of its encoded P2 protein were analyzed. The P2 protein was found to share low sequence similarity with other badnaviruses and elicit a hypersensitive-like response, and suppress post-transcriptional and transcriptional gene silencing, suggesting that P2 is a viral pathogenicity factor. Our findings increase the understanding of the pathogenesis of SCBV virus and lay a foundation for antiviral resistance breeding.

Materials and Methods

Source of plant materials

Sugarcane plants of the Badila cultivar that exhibited mottling, stunting, and chlorotic streaking symptoms were collected from Menghai County, Yunnan Province, China, and rapidly frozen in liquid nitrogen and then stored at −80 °C.

Multiple sequence alignment and phylogenetic analyses

Amino acid sequence similarities were determined for the P2 of six representative members of the genus Badnavirus and the sole member of Tungrovirus (Rice tungro bacilliform virus, RTBV); all sequences and the corresponding GenBank accession numbers are listed in Table 1. Sequence multiple alignment was performed with the Clustal Omega method using the SnapGene software. The corresponding sequences of the above viruses were aligned, and percent identities were determined in Clustal W. The phylogenetic tree was inferred via the Neighbor-joining method in MEGA11 with the bootstrap 1,000 replicates.

Table 1 Amino acid sequences of P2 from Sugarcane bacilliform virus (SCBV) and representatives within genus Badnavirus and Tungrovirus.

Virus a	Genus	GenBank accession number	Size (aa)	Amino acid sequence identity (%)	
SCBV	Badnavirus	WON00947	123	–	
BSOLV	Badnavirus	NP_569149	135	26.8	
CLNV	Badnavirus	YP_002117530	128	31.1	
CoYMV	Badnavirus	NP_039819	136	19.5	
CSSV	Badnavirus	NP_041733	133	24.6	
PYMoV	Badnavirus	AXG50781	159	23.8	
TaBCHV	Badnavirus	YP_009130663	126	28.3	
RTBV	Tungrovirus	AAD30189	111	22.9	
Note:

a BSOLV, Banana streak OL virus; CLNV, Cycad leaf necrosis virus; CoYMV, Commelina yellow mottle virus; CSSV, Cacao swollen shoot virus; PYMoV, Piper yellow mottle virus; RTBV, Rice tungro bacilliform virus; SCBV, Sugarcane bacilliform virus; TaBCHV, Taro bacilliform CH virus.

Construction of plasmids

Total genomic DNA was extracted using cetyltrimethylammonium bromide (CTAB)-based methods as previously described (Springer, 2010). Genomic DNA was purified after RNase A digestion and used as a template for PCR. The complete genome of SCBV was obtained by isothermal amplification with Phi29 MAX DNA Polymerase (Cat#: N106-01; Vazyme, Nanjing, China) according to the manufacturer’s instructions. The products of isothermal amplification were then used as templates for amplification of the complete genome using primer pair SCBV/F and SCBV/SmaI/R (Table S1) and Phanta EVO HS Super-Fidelity DNA Polymerase (Cat#: P504-d1; Vazyme, Nanjing, China). The SCBV complete genome sequence was then subcloned into the pCE2-TA/Blunt-Zero vector (Cat#: C601-01; Vazyme, Nanjing, China) and transformed into Escherichia coli Top10. The positive colonies were cultured and verified by PCR and Sanger sequencing. The complete genome of SCBV was then submitted to the National Center for Biotechnology Information (GenBank accession number: OR672147). The ORF2 coding sequence was amplified by PCR and cloned into PVX-based vector pGR106 by digested with ClaI and SalI followed by T4 DNA ligase (Cat#: EL0014,ThermoFisher Scientific, Shanghai, China) ligation to get the recombinant plasmid PVX-P2. Also, P2 was inserted into the pCHF3 binary expression vector or fused to the N-terminal of enhanced green fluorescent protein (eGFP) in pCHF3-eGFP by double enzyme digestion with SacI/BamHI. The corresponding recombinant plasmids were referred to as pCHF3-P2 and pCHF3-P2-eGFP, respectively. All primers used in this study are listed in Table S1.

Plant growth and agroinoculation

Wild-type, 16c (Voinnet & Baulcombe, 1997), 16-TGS (Raja et al., 2008) and RFP-H2B transgenic seedlings of N. benthamiana were grown to the 4- to 5-leaf stage in an insect-free chamber at a constant temperature of 25 °C and a 16-h/8-h day/night cycle. The binary plasmids pCHF3-P2 and empty pCHF3 were transfected into Agrobacterium tumefaciens strain EHA105, and the recombinant plasmids PVX and PVX-P2 were transformed into A. tumefaciens GV3101 by electroporation. Suspensions of A. tumefaciens cultures were regulated to OD600 of 1.0, and the Agrobacterium containing PVX-P2 was infiltrated into N. benthamiana wild-type or 16-TGS seedlings using a 1-mL syringe without a needle. Transient PTGS suppression experiments were conducted as previously described (Johansen & Carrington, 2001; Li et al., 2014, 2015).

Plant transformation

The transgenic N. benthamiana plants over-expressing P2 or empty vector were generated by using the Agrobacterium-mediated leaf disc transformation method. The binary empty vector pCHF3 or recombinant plasmid pCHF3-P2 were transformed into A. tumefaciens strain EHA105 and used for transfection of N. benthamiana leaf discs. Potential transformants were selected on MS media containing 200 μg ml−1 cefotaxime and 200 μg ml−1 kanamycin. Kanamycin-resistant cluster buds were cut off, placed on rooting media, cultured to a height of 5–6 cm, and then transplanted into soil. Transgenic seedlings were verified by PCR with CaMV 35S promoter or P2-specific primers, respectively. Relative levels of P2 mRNA in transgenic plants were confirmed by RT-qPCR.

H2O2 detection in plants

H2O2 production was detected visually in N. benthamiana leaves using the 3,3′-diaminobenzidine (DAB) staining method (Cat#: A690009; Sangon Biotech, Shanghai, China) (Sharma & Ikegami, 2010) and making some modifications as previously described (Liang et al., 2023).

Subcellular localization analysis

For subcellular localization experiments, fluorescence in RFP-H2B transgenic N. benthamiana leaf epidermal cells or protoplasts inoculated with pCHF3-eGFP and pCHF3-P2-eGFP was examined by confocal microscopy (Leica TCS SP8MP; Leica, Mannheim, Germany) 2- to 3-days post inoculation (dpi) as described (Shen et al., 2011; Yoo, Cho & Sheen, 2007).

Protoplast preparation

To further observe the more precise subcellular localization of SCBV P2 protein, the protoplasts of RFP-H2B transgenic N. benthamiana leaf epidermal cells inoculated with pCHF3-eGFP or pCHF3-P2-eGFP were prepared by digesting with 1.5% (wt/vol) cellulase (Cat#: A002610; Sangon Biotech, Shanghai, China) and 0.4% (wt/vol) macerozyme R10 (Cat#: A004297; Sangon Biotech, Shanghai, China) as described (Yoo, Cho & Sheen, 2007).

DNA methylation analysis by restriction digestion

The genomic DNA of pCHF3 and P2 transgenic plants was extracted using the CTAB method. Digestion analysis of genomic DNA was performed by using a methylation-insensitive restriction endonuclease BamHI (Cat#: FD0054; Thermo Fisher Scientific, Shanghai, China) or a methylation-dependent endonuclease McrBC (Cat#: M0272; New England Biolabs, Ipswich, MA, USA), respectively. The restriction digestion reaction system (50 μL) consists of 10 μg of genomic DNA and 50 U of the respective endonuclease according to the manufacturer’s specifications. Digested products were instantly separated by electrophoresis through a 1.5% agarose gel.

RT-qPCR

RT-qPCR analysis was performed to measure the transcription level of core genes in the RNA-directed DNA methylation (RdDM) pathway. Total RNAs from PVX-inoculated N. benthamiana and P2 or empty vector-transgenic N. benthamiana plants were extracted as experimental and control groups using RNAiso Plus reagent (Cat#: 9108, Takara, Beijing, China) and the A260/A280 value and concentration of RNA products were measured by NanoPhotometer® N60/N50 (Implen, Munich, Germany). A total of 2 μg of high-quality RNA were converted to cDNA using HiScript III 1st Strand cDNA Synthesis Kit (+gDNA wiper) (Cat#: R312-01; Vazyme, Nanjing, China) according to the manufacturer’s instructions. The cDNA product was diluted 10-fold and served as a template for RT-qPCR. Each reaction mixture contained 2 μL of the diluted cDNA, 10 μL of 2 × ChamQ Universal SYBR qPCR Master Mix (Cat#: Q711-02; Vazyme, Nanjing, China), 0.4 μL of each of the forward and reverse primer (10 μM) in a total volume of 20 μL (primers are listed in Table S1). Three independent biological and experimental replicates were performed, and the reactions were run under the following program conditions: 95 °C for 30 s, 40 cycles of 95 °C for 10 s, and 60 °C for 30 s. Melting curves were derived (95 °C for 15 s, 60 °C for 60 s, and 95 °C for 15 s) for each reaction to ensure a single product. Reactions were performed using the LightCycler 96 real-time PCR system (F. Hoffmann-La Roche Ltd., Basel, Switzerland). The qPCR data was analyzed using the 2−ΔΔCt method (Livak & Schmittgen, 2001). All primers used for qRT-PCR detection in this study are listed in Table S1.

Immunoblotting

As described previously, the protein of systemic leaves was extracted from PVX-infected plants (Xiong et al., 2009). Immunoblotting was performed with primary Mouse anti-PVX CP monoclonal antibodies, followed by HRP-conjugated Goat Anti-Mouse IgG (Cat#: D110087, Sangon Biotech, Shanghai, China). Blotted membranes were washed thoroughly and visualized using the SuperPico ECL Chemiluminescence Kit according to the manufacturer’s protocol (Cat#: E422; Vazyme, Nanjing, China).

Results

Phylogenetic relationships between SCBV P2 and other taxa of the genus Badnavirus

In this study, five sugarcane samples were collected and detected by RT-PCR; two samples were SCBV positive. One isolate’s complete genome of SCBV was sequenced and submitted to NCBI (GenBank accession number: OR672147). The ORF2 of SCBV is 372 nucleotides (nt) long and encodes a small protein of 123 amino acids (aa) called P2. The complete amino acid sequences of SCBV P2, Rice tungro bacilliform virus (RTBV, GenBank accession number: AAD30189), and several representative species of the genus Badnavirus (as seen in Table 1) were aligned and phylogenetic analyzed. Multiple sequence alignment shows that all P2 amino acid sequences have no obvious sequence similarity (Fig. 1A, Table 1). A coiled-coil like domain (47-LLTLHGKITALEGRLQDLKEDIAKKADK-74) was predicted in SCBV P2 by InterPro online prediction (https://www.ebi.ac.uk/interpro/). The coiled-coil like domain is relatively conserved in length and position in all the above viruses (Fig. 1A). Phylogenetic analysis shows that all the amino acid sequences of P2 proteins from eight individual viruses were clustered into three groups. SCBV P2 was assigned in a clade alone and had a relatively closer relationship with Banana streak OL virus and Cycad leaf necrosis virus P2, and these three viruses were clustered in the same group (Fig. 1B).We have previously demonstrated that SCBV P2 can bind to both homologous and heterologous nucleic acids in a sequence-nonspecific manner, and the coiled-coil like domain plays a vital role in P2-nucleic acids binding through self-interaction (Lou et al., 2023), and this is consistent with that of CSSV and RTBV (Jacquot et al., 1996; Jacquot, Keller & Yot, 1997). The above results suggest that although the badnaviruses P2 share little sequence similarity, they all contain a conserved coiled-coil like domain, which plays an indispensable role in P2-nucleic acids affinity binding. This property is conserved in Badnavirus and Tungrovirus.

Figure 1 Phylogenetic relationships between SCBV and other taxa in the genus Badnavirus and Tungrovirus.

(A) Multiple alignment of amino acid sequence of SCBV P2 and representatives of Badnavirus and Tungrovirus. The conserved amino acid residues were highlighted in yellow, and the coiled-coil like motifs were framed in a black rectangle. (B) Unrooted neighbor-joining phylogenetic tree reconstructed from the alignment of the amino acid sequences of SCBV P2 and other taxa in the genus Badnavirus and Tungrovirus. The phylogenetic tree was constructed by using the MEGA11 program and the percentage of bootstrap values (1,000 replicates) are shown at the branch internodes. BSOLV, Banana streak OL virus; CLNV, Cycad leaf necrosis virus; CoYMV, Commelina yellow mottle virus; CSSV, Cacao swollen shoot virus; PYMoV, Piper yellow mottle virus; RTBV, Rice tungro bacilliform virus; SCBV, Sugarcane bacilliform virus; TaBCHV, Taro bacilliform CH virus.

Subcellular localization of the SCBV P2 protein

To determine the precise subcellular localization of SCBV P2, an enhanced green fluorescent protein (eGFP) was fused to the C terminus of P2 (P2-eGFP) and subcloned into the binary expression vector pCHF3 under the transcription of the Cauliflower mosaic virus 35S promoter. A. tumefaciens containing the recombinant plasmid pCHF3-P2-eGFP or the pCHF3-eGFP vector were infiltrated into 4- to 5-leaf stage RFP-H2B transgenic N. benthamiana plants (containing an RFP nucleus localization signal), respectively. Green fluorescence in inoculum leaves was observed at 2- to 3-dpi, using confocal microscopy. In plants expressing eGFP alone (35S-eGFP), the fluorescence was observed in the cytoplasm and the nucleus. Similarly, the fluorescence in P2-eGFP-infiltrated leaves was also found in the nucleus as well as cytoplasm (Fig. 2A). To further confirm the subcellular localization of P2, the inoculated leaves were digested with cellulase and macerozyme to obtain the protoplasts. The eGFP signal of 35S-eGFP infiltrated mesophyll protoplasts was distributed around the periphery of the cytomembrane and nucleus, and it is similar in the 35S-P2-eGFP infiltrated mesophyll protoplasts except for some dense bright fluorescent spots (Fig. 2B) The above results indicate P2 have a cytoplasm and nucleus subcellular co-localization.

Figure 2 Subcellular localization of SCBV P2 in RFP-H2B transgenic Nicotiana benthamiana mesophyll cells and protoplasts.

(A) Subcellular localization of P2 fused to the enhanced green ûuorescent protein (eGFP) in RFP-H2B transgenic N. benthamiana mesophyll cells. (B) Subcellular localization of P2 fused to eGFP in RFP-H2B transgenic N. benthamiana protoplasts. The 35S-eGFP expression plasmid was used as a control. The ratio scale was shown in the bottom right corner of the picture.

RNA silencing suppressor activity of SCBV P2

RNA silencing is an efficient innate antiviral mechanism possessed by plants (Li & Wang, 2019). To repress the transcription of viral DNAs, host Dicer-like protein directs TGS through RdDM, or PTGS, which includes splicing and degradation or translational repression of recognized viral RNA (Boualem, Dogimont & Bendahmane, 2016; Matzke & Mosher, 2014). To date, the badnaviruses encoded RNA silencing suppressor (RSS) has been rarely reported. As we described previously that SCBV P2 can bind both DNA and RNA (Lou et al., 2023) and is localized in both the cytoplasm and nucleus, suggesting the possibility that it could function as a viral RSS and support viral infection. To test this hypothesis, we used a 16c-transgenic N. benthamiana line as the experimental plant, which can constitutively express a green fluorescent protein (GFP) signal localized in the endoplasmic reticulum (ER). A. tumefaciens harboring pCHF3-P2 or pCHF3 (negative control) and the p19 suppressor encoded by Tomato bushy stunt virus (TBSV) (positive control) was individually mixed with an equal volume of A. tumefaciens containing a recombinant plasmid expressing the RNA silencing inducer 35S-GFP and inoculated into 4- to 5-leaf aged 16c-transgenic N. benthamiana seedlings. By 4 dpi, the strength of green fluorescence in leaves infiltrated with the pCHF3 empty vector had decreased dramatically under UV light, and was almost undetectable under stereo fluorescence microscope. However, the intensity remained relatively high in patches expressing P2, and those inoculated with p19 had the highest fluorescence intensity (Fig. 3A), these data indicate that SCBV P2 is an RSS, which can suppress single stranded RNA (ssRNA) triggered local gene silencing.

Figure 3 SCBV P2 inhibits ssRNA induced PTGS and reverses TGS.

(A) Repression of GFP silencing in N. benthamiana 16c leaves. Leaf areas were co-infiltrated with A. tumefaciens expressing GFP (35S-GFP) and either a pCHF3 control, SCBV P2 (pCHF3-SCBV-P2), or TBSV p19 (pCHF3-p19). Photos of the above infiltrated leaves were taken at 4 dpi, under high intensity UV light. (B) Leaf patches of N. benthamiana were co-infiltrated with A. tumefaciens carrying GFP (35S-GFP) and dsFP (35S-dsFP), as well as pCHF3 vector, SCBV P2 or p19, and typical leaf patches were photographed under high intensity UV light at 4 dpi. (C) Plant of N. benthamiana 16-TGS were left uninoculated, or inoculated with PVX, PVX-P2, or PVX- β C1, respectively, and the apex of plants and leaves were photographed under high intensity UV light at 21 dpi.

To test whether SCBV P2 can suppress dsRNA-induced gene silencing, wild-type N. benthamiana plants of 4- to 5-leaf age were infiltrated with A. tumefaciens solutions containing the same volume of transient dsRNA elicitor 35S-GFP and 35S-dsFP together with pCHF3, pCHF3-P2, and pCHF3-p19, respectively. As shown in Fig. 3B, leaf spots infiltrated with neither pCHF3 nor pCHF3-P2 mixed with the silencing inducer did not show any GFP fluorescence at 4 dpi under UV light or stereo fluorescence microscopy, indicating that P2 fails to suppress dsRNA-induced RNA silencing. In contrast, a high intensity of green fluorescence was observed in patches expressing the p19 positive control (Fig. 3B). The above data suggest that SCBV P2 is a weak local ss-PTGS but not a ds-PTGS suppressor.

To determine whether P2 can also conquer TGS, a transgenic N. benthamiana line named 16-TGS (the CaMV 35S promoter of the GFP transgene is transcriptionally silenced) was used. Transgenic seedlings of 4- to 5-leaf stage were inoculated with GV3101 A. tumefaciens solutions (mock) or PVX-P2, PVX (negative control), or PVX-βC1 (the TGS repressor of Tomato yellow leaf curl China betasatellite (TYLCCNB), positive control), respectively. After 21 dpi, PVX-infected plants were almost asymptomatic, and no visible green fluorescence was detected in systemic tissues under a high-intensity UV lamp. However, PVX-βC1-infected plants exhibited severe dwarfing, stem deformation, and upward leaf curling along with the PVX-immanent mosaic symptoms, and GFP fluorescence was quite noticeable. As shown in Fig. 3C, systemic leaves infected with PVX-P2 showed severe mosaic and spotty mottling symptoms accompanied by visible green fluorescence under UV light (Fig. 3C), indicating that P2 is a potential TGS repressor.

SCBV P2 expression impacts RdDM signaling in the host

In plants, DNA methylation is a conserved epigenetic modification that regulates genome stability, gene expression, and antiviral defense (He et al., 2022; Wang et al., 2019; Zhang, Lang & Zhu, 2018). Moreover, the methylation level of the plant host is reprogrammed when challenged by an invading virus. To investigate the effect of P2 on the global methylation patterns of the host plant, we constructed a transgenic P2 line of N. benthamiana and tested methylation at the genome level using a methylcytosine-dependent endonuclease (McrBC). Genomic DNA from pCHF3 (Vec) and P2 transgenic N. benthamiana plants (5# and 9#) were extracted and subsequently processed with restriction digestion assays. A mock treatment without enzyme was performed as a control (Sham), and all genomic DNA samples remained unaltered. When a methylation-insensitive endonuclease (BamHI) was used, we found that all three genomic DNA samples were digested and formed a ‘smear’ pattern on the agarose gel during electrophoresis (Fig. 4A). However, McrBC treatment cleaved a portion of the genomic DNA from Vec transgenic lines. In the contrary, the DNA from the P2 transgenic lines showed high resistance and remained unchanged (Fig. 4A). The present data suggest that P2 can reduce DNA methylation on a genome-wide scale when transgenically expressed in N. benthamiana plants.

Figure 4 SCBV P2 expression impacts host RdDM pathway andgenome-wide scale methylation.

(A) DNA methylation analysis in P2 transgenic N. benthamiana plants using restriction endonuclease digestion. The methylation-dependent endonuclease McrB. and the methylation-insensitive enzyme BamHI were used to digest genomic DNA isolated from the vector control (Vec) and two separate lines of P2 transgenic plants (5# and 9#). The term ‘Sham’ refers to a simulated digestion that contains no enzyme. The positions of the uncut input and the digested products are shown. (B) SCBV P2 Overexpression inhibits transcription of N. benthamiana ARGONAUTE 1 (NbAGO1) and NbAGO4 in PVX-treated plants. RT-qPCR assays were performed to analyze the effects of P2 on the expression of homologous genes of DNA methyltransferases, demethylases, histone deacetylase and essential genes related to RdDM. Relative expression levels of DNA METHYLTRANSFERASE1 (NbMET1) (GenBank accession number: FJ222441), DOMAINS REARRANGED METHYLTRANSFERASE2 (NbDRM2) (JQ957857), CHROMOMETHYLASE3 (NbCMT3) (JQ957858), DICER3 (NbDCL3) (FM986782), REPRESSOR OF SILENCING 1 (NbROS1) (JQ957859), NbROS2 (JQ957860), NbAGO1 (DQ321488), NbAGO4 (DQ321490) and Histone Deacetylase 6 (NbHDA6) (KU170188) were measured in PVX and PVX-P2 inoculated N. benthamiana plants at 15 dpi (B), or in pCHF3 vector and P2 transgenic plants at 30 days after sprouting (C). T-tests were performed to analyze the significance of difference (*P < 0.05, **P < 0.01). Each of the experiments were carried out at least three times.

To explore the possible mechanism of SCBV P2 in repressing epigenetic TGS and genome-wide DNA methylation, the relative expression levels of the homologous genes of DNA methyltransferases, demethylases, histone deacetylase and essential genes related to the RdDM pathway were analyzed in P2 or the empty vector transgenic N. benthamiana plants. Specific primers were designed and synthesized for qRT-PCR detection of the homologs of DNA methyltransferases (MET1, DRM2, and CMT3), demethylases (ROS1, ROS2), argonautes (AGO1, AGO4), dicers (DCL3), and histone deacetylase 6 (HDA6). Total RNA from the empty vector (Vec) and P2 transgenic N. benthamiana plants was extracted 4 weeks after sowing, and total RNA from PVX- or PVX-P2-infected N. benthamiana plants was isolated at 15 dpi. The above RNA was reverse transcribed into cDNA and serves as templates for the subsequent RT-qPCR assays. As shown in Figs. 4B and 4C, the expression of NbAGO1 was significantly down-regulated in both P2 transgenic plants and PVX-P2-infiltrated plants, and the expression of NbAGO4 was dramatically reduced in P2 transgenic N. benthamiana plants but remained inconspicuous in PVX-P2-inoculated plants (Fig. 4B and 4C). Surprisingly, we found that the expression level of NbHDA6 was also significantly down-regulated in both P2 transgenic plants and PVX-P2-infiltrated plants. Further, the HDA6 protein has been proven to act as a histone deacetylase and cofactor of MET1, which stimulates DNA methylation. Taken together, we speculate that constitutive overexpression of SCBV P2 may suppress host TGS by inhibiting the expression of NbAGO1 and NbHDA6.

SCBV P2 induces a hypersensitive-like response (HLR) in N. benthamiana

SCBV infection usually causes patchy, chlorotic streaking symptoms (Viswanathan, Alexander & Garg, 1996) and induces broken chlorotic streaks in inoculated rice (Bouhida, Lockhart & Olszewski, 1993). We successfully constructed an infectious clone of SCBV by inserting a 1.06 copy of the tandem viral genome into the binary vector pCB301-2×35S-HDVRZ-NOS (Figs. S1A and S1B). After Agrobacterium-mediated inoculation, SCBV-infected rice seedlings showed stunting and broken chlorotic streaks on leaves at 21 dpi (Fig. S1C), consistent with the results of Bouhida and colleagues (Bouhida, Lockhart & Olszewski, 1993). We also investigated whether the necrotic streaks were the result of a hypersensitive-like response. The rice leaves that showed typical necrotic streaks were stained with 3,3′-diaminobenzidine (DAB), and the mock and pCB301-2×35S-HDVRZ-NOS empty vector infiltrated plants were used as controls. As shown in Fig. S1, brown necrotic spots appeared in leaves infected with SCBV, whereas plants inoculated with mock and empty vector remained transparent and speckless. The accumulation of viral genomic DNA was verified by PCR analysis, and the results indicate that infectious SCBV can successfully infect rice and cause H2O2 accumulation-induced HLR.

To evaluate whether SCBV P2 is a symptom elicitor in N. benthamiana, the P2 protein was ectopically overexpressed by a PVX-based vector. N. benthamiana plants of 4- to 5-leaf stage were inoculated with A. tumefaciens harboring PVX or PVX-P2, respectively. At 7 dpi, PVX-infected plants began to show typical mosaic and shriveling symptoms, whereas PVX-P2-infiltrated plants remained symptomless (Fig. 5A, upper panels). At 10 dpi, the symptoms of PVX-P2-infected plants resembled those of PVX-infected plants (Fig. 5A, middle panels). Symptoms of PVX-infected plants showed signs of recovery from 12 dpi, and vein chlorosis and mosaic phenotype disappeared by 20 dpi, whereas P2-expressing plants still exhibited mosaic and leaf wrinkling phenotypes (Fig. 5A, lower panels). These findings suggest that SCBV P2 can delay the onset of host symptoms to some extent and persist for a considerably longer period of time. To demonstrate the relationship between necrosis and H2O2 accumulation, the 3,3′-diaminobenzidine (DAB) staining assay was performed to detect the accumulation of H2O2 in empty or PVX-P2 inoculated plants at 10 dpi and 20 dpi. We found that P2-expressing leaves exhibited many brown necrotic spots at 20 dpi, whereas leaves inoculated with the vector remained spotless (Fig. 5B). Protein immunoblotting analyses confirmed that PVX CP accumulated in more significant amounts in PVX-P2-inoculated plants than that in PVX-inoculated plants (Fig. 5C), suggesting that P2 is a potential virulence factor that promotes PVX replication and accumulation. These results indicate that SCBV P2 protein is a viral pathogenicity factor that can induce a hypersensitive-like response and promote virus accumulation.

Figure 5 Symptoms of the plants inoculated with Potato virus X(PVX) or PVX-P2.

(A) Symptoms evoked on N. benthamiana plants at 7-, 10- and 20-dpi inoculated with PVX or PVX-P2. (B) PVX-P2 induced mosaic symptoms and necrotic spots on N. benthamiana plants. Upper systemic leaves were photographed at 10- and 20-dpi, respectively. Followed by photographing after 3,3′-diaminobenzidine (DAB) treatment. The necrotic lesions are shown by the red arrowheads. (C) Western blot analysis of coat protein (CP) accumulation in PVX or PVX-P2 infected plants. Freshly emerging leaves were used to extract total protein. The anti-PVX CP monoclonal antibody was used to detect the accumulation of PVX, Coomassie light blue stained Rubisco large subunit protein was used as a loading control. The gray values of the blot bands were evaluated by using the ImageJ software, the relative amount of CP accumulation in PVX-infected plants was preset as 100%.

Discussion

Sugarcane bacilliform virus (SCBV) is an important member of Badnavirus that causes severe quality and yield losses worldwide. Although numerous efforts have been made to reveal the molecular characteristics, pathogenicity, and pathogenesis of SCBV, little is clear. We have previously demonstrated that the P2 protein of SCBV could bind both viral-derived and heterogenous DNA or RNA in a sequence nonspecific manner (Lou et al., 2023). This observation is consistent with the findings of previous studies conducted on CSSV, CoYMV, and RTBV. (Cheng, Lockhart & Olszewski, 1996; Jacquot et al., 1996; Jacquot, Keller & Yot, 1997). Although the P2 proteins of badnaviruses do not show apparent sequence homology (Fig. 1), they all possess nucleic acid binding activity, which appears to be a universal property of badnaviruses P2 proteins. Furthermore, we have proved that the coiled-coil like region in P2 is critical for self-interaction and nucleic acid binding (Lou et al., 2023), and the coiled-coil like region is highly conserved in size and position among Badnavirus and Tungrovirus, indicating its universal crucial role for the P2 protein. Since P2 has the ability to bind nucleic acids, it could play a key role in preserving viral nucleic acids from degradation or being involved in virion assembly.

RNA silencing is an evolutionarily conserved immune barrier of host plants in defending microbes, such as viruses (Li & Wang, 2019). To overcome this defense, viruses have evolved a variety of proteins (such as RSSs) capable of suppressing host gene silencing by other PTGS or TGS via the RdDM pathway (Boualem, Dogimont & Bendahmane, 2016; Matzke & Mosher, 2014). In general, viral-encoded RSSs are multifunctional and can play critical roles in various stages of the virus infection in addition to suppressing RNA silencing (Csorba, Kontra & Burgyán, 2015; Yang & Li, 2018). For example, CaMV P6 has been proven to act as an RSS by suppressing the activity of DRB4 (Haas et al., 2015). RTBV P4 can inhibit the production of siRNA to suppress RNA silencing (Rajeswaran et al., 2014). Here, we tested the PTGS repressor activity of SCBV P2 and found that P2 can inhibit sense RNA-induced but not dsRNA-induced PTGS (Fig. 3), implying that P2 may also repress the formation of dsRNA or degradation but has no effect on siRNA metabolism. In fact, some other Caulimovirus-derived suppressor proteins, such as P6 encoded by Strawberry vein banding virus (SVBV), may interfere with dsRNA degradation and act as an RSS (Feng et al., 2018).

RdDM is a common epigenetic modification that plays critical roles in gene expression regulation and defense against invading viruses. As a counter-defense strategy, some plant viruses encode TGS suppressors as a tactic to block the activity of essential enzymes or protect the substrates from degradation in the subsequent methylation cycle (Ismayil et al., 2018; Raja et al., 2008; Yang et al., 2011; Zhang et al., 2011). For example, the HC-Pro protein from the Potyviridae family is involved in lots of processes of RNA silencing suppression by blocking methylation of the 3′-end of siRNA or directly binding to AGO1 and downregulating its expression, etc., (Valli et al., 2018). Moreover, the P0 proteins from the genus Polerovirus have been shown to block the binding of siRNAs/miRNAs to the free AGO effector to form an intact RNA-induced silencing complex (RISC) and mediate the degradation of AGO1 via the autophagy pathway (Csorba et al., 2010; Michaeli et al., 2019). In this study, we verified that SCBV P2 inhibits TGS by suppressing the expression of AGO1 and AGO4 (Figs. 4B and 4C), which were known to defend against plant RNA viruses (Morel et al., 2002; Wu et al., 2015; Zhang et al., 2006) and DNA viruses (Raja et al., 2008), respectively. AGO1 regulates gene expression in a variety of developmental and physiological processes (Fei, Xia & Meyers, 2013; Rogers & Chen, 2013). It also functions in virus defense when loaded with viral siRNAs via dsRNA-triggered gene silencing (Fang & Qi, 2016; Wu et al., 2015), and in this study, P2 downregulates the expression level of NbAGO1 but has no visible influence on ds-PTGS (Figs. 3 and 4), although the details remain to be elucidated. The AGO4 protein is a crucial component of the RdDM pathway that recruits DRM2, a major de novo methyltransferase, to add methyl to target DNA (Cao & Jacobsen, 2002; Zhong et al., 2014). Furthermore, we found that SCBV P2 transgenic plants showed lower levels of methylation genome-wide than empty vector transgenic seedlings (Fig. 4A). Taken together, these results provide a model that SCBV P2 represses host DNA methylation by suppressing or disrupting various components involved in the DNA methylation pathway.

Oxidative burst (including O−2 and H2O2) induced HR (Mubin et al., 2010) is universal when plants respond to pathogens and is thought to limit pathogen growth or movement (Hussain et al., 2007; Lam, Kato & Lawton, 2001). In this study, we found that SCBV P2 could induce typical mosaic symptoms and HLR necrosis in the late stage of infection with a PVX-based vector (Fig. 5). This suggests that P2 somewhat delays the onset of viral symptoms but triggers H2O2 accumulation and causes necrosis symptoms. These results are consistent with the findings of inoculation with infectious clones in Oryza sativa (Fig. S1), further confirming that P2 is the pathogenic factor encoded by SCBV. Intriguingly, several plant DNA viruses encoding TGS suppressors have been verified to be virulence factors inducing HR, such as the V2 protein of Papaya leaf curl virus (PaLCuV), Cotton leaf curl Kokhran virus (CLCuKoV) and Tomato leaf curl Java virus (ToLCJV) (Hussain et al., 2007; Mubin et al., 2010; Sharma & Ikegami, 2010). Based on the above results, we suppose that the pathogenesis and TGS repressor activity of SCBV P2 are coupled. Altogether, these observations indicate that SCBV P2 is a multifunctional protein that can suppress PTGS and TGS as well as induce HLR. All these findings help to elucidate the molecular pathogenesis of Badnavirus and provide a possible target for future antiviral breeding.

Conclusions

Our work reveals the fact that the P2 protein encoded by Sugarcane bacilliform virus (SCBV) plays a vital role in the pathogenicity of the virus, which serves as a ss-PTGS suppressor and represses host TGS by inhibiting core genes transcription in RdDM pathway, such as AGO1. In addition, P2 was proven to be a virulence factor that can induce HLR and assist PVX accumulation in N. benthamiana. Our conclusions increase the awareness of the molecular mechanism of the pathogenesis of SCBV and help lay a foundation for disease resistance breeding.

Supplemental Information

Supplemental Information 1 Primer sequences used in this study.

Supplemental Information 2 The construction of SCBV infectious clone and symptomsinduced on Oryza sativa plants.

(A) The complete genome struct of SCBV. (B) The construction strategy of SCBV infectious clone. (C) Symptoms elicited on Oryza sativa plants at 21 dpi infected with SCBV infectious clone and pCB301 empty vector (a-f). Necrotic lesions on Oryza sativa leaves induced by SCBV were photographed directly at 21 dpi and photographed after 3,3′-diaminobenzidine (DAB) staining (g-i). (D) SCBV detection on SCBV inoculated Oryza sativa plants at 21 dpi by PCR using RT/RNase H region specific primer pairs.

Supplemental Information 3 Raw data of Figure 1.

Supplemental Information 4 Raw data of Figure 2.

Supplemental Information 5 Sequence data of SCBV.

Supplemental Information 6 Raw data of the IR-PTGS suppressor activity of SCBV P2 in Figure 3.

Supplemental Information 7 Raw data of the SS-PTGS suppressor activity of SCBV P2 in Figure 3.

Supplemental Information 8 Raw data of the TGS suppressor activity of SCBV P2 in Figure 3.

Supplemental Information 9 Raw data of Figure 4.

The uncropped gels of restriction endonuclease digestion assays and raw data of Real-Time Quantitative PCR.

Supplemental Information 10 The symptoms caused by SCBV P2 in Figure 5-trial 1.

Supplemental Information 11 The symptoms caused by SCBV P2 in Figure 5-trial 2.

Supplemental Information 12 The symptoms caused by SCBV P2 in Figure 5-trial 3.

Supplemental Information 13 Raw data of DAB staining in Figure 5.

Supplemental Information 14 Raw data of the Western blot and Coomassie light blue staining in Figure 5.

Supplemental Information 15 Raw data of the symptoms caused by SCBV infectious clone in Rice (of Figure S1-1).

Supplemental Information 16 Raw data of DAB staining of rice leaves in Figure S1-2.

Supplemental Information 17 MIQE Checklist.

We thank professor Xueping Zhou from Institute of Plant Protection, Chinese Academy of Agricultural Sciences for kindly giving the PVX-based expression vector and Mouse anti-PVX CP monoclonal antibody and the revision and correction of the whole grammar. We also thank professor Xiaorong Tao from Nanjing Agriculture University for kindly providing the pCB301-2×35S-HDVRZ-NOS binary vector.

Additional Information and Declarations

Competing Interests

Author Contributions

DNA Deposition

Data Availability

The authors declare that they have no competing interests.

Xiongbiao Xu conceived and designed the experiments, analyzed the data, authored or reviewed drafts of the article, and approved the final draft.

Yinian Lou performed the experiments, analyzed the data, prepared figures and/or tables, and approved the final draft.

Kaili Liang performed the experiments, analyzed the data, prepared figures and/or tables, and approved the final draft.

Jingying Liu performed the experiments, analyzed the data, prepared figures and/or tables, and approved the final draft.

Zhiyuan Wang performed the experiments, analyzed the data, prepared figures and/or tables, and approved the final draft.

Baoshan Chen conceived and designed the experiments, authored or reviewed drafts of the article, and approved the final draft.

Wenlan Li conceived and designed the experiments, authored or reviewed drafts of the article, and approved the final draft.

The following information was supplied regarding the deposition of DNA sequences:

The sugarcane bacilliform virus isolate MH2, complete genome

is available at GenBank: OR672147.

The following information was supplied regarding data availability:

The raw measurements are available in the Supplemental File.

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
