# Peer review of "The P2 nucleic acid binding protein of Sugarcane bacilliform virus is a viral pathogenic factor"

_PeerJ, doi:10.7717/peerj.16982_

## Round 0.1 · original submission · Minor Revisions

Please address the suggestions of reviewer 2. In particular, supply further experimental evidence for the influence of P2 on DNA methylation or tone down your conclusions in respect to the role of P2 in DNA-methylation.

In addition, check the annotations of reviewer 1 in the script. Add the missing information and the requested references.

**Language Note:** The review process has identified that the English language must be improved. PeerJ can provide language editing services - please contact us at copyediting@peerj.com for pricing (be sure to provide your manuscript number and title). Alternatively, you should make your own arrangements to improve the language quality and provide details in your response letter. – PeerJ Staff

·

Basic reporting

Every thing is fine.

No Comment

Experimental design

Every thing is fine.

No Comment

Validity of the findings

Every thing is fine.

No Comment

Additional comments

Scientific name must be in Italics

Check the references as per the Journal format

Check the Figure Numbers

Reviewer 2 ·

Basic reporting

The aim of the research study of Xu et al. is to investigate the function of the SCBV protein P2 in viral infection. Different experimental approaches are employed to elucidate the role of P2 in TGS, PTGS and the induction of HR. Xu et al. conclude P2 to have a role as a pathogenicity factor, in particular through silencing-suppression.

The language employed in this article needs some improvement in order to make the work more comprehensible. This includes consistency in tenses (line 157) clarity and correct technical terms (line 59, 191, 195, 205, 214, 293-294, 334, 343, and more). Sentences of excessive length (lines 236-240) should be avoided. I thus recommend revision of the text by a native or proficient English speaker.

The research has been put into context with respect to the current state of knowledge; however, certain parts need to be clarified in order to improve understandability:
Lines 72-75: I assume that the fragment (ORF2) is between 3’-end of ORF1 & 5’-end of ORF3; relate ORF2 to its gene product, P2.
Lines 74-78: Introduce the two example viruses (CoYMV, CSSV) as badnaviruses.
Lines 79-80/189-190: Clarify the connection between RTBV (genus Tungrovirus) and Badnaviruses and why this example was included in this study.
Lines 196-197: Please clarify if it is the viruses or the 8 viral proteins that were clustered.

Literature: The article from Lou et al., 2023, is only available in Chinese. That makes it hard to evaluate the role of the coiled-coil domain which is claimed to be present in SCBV P2.

Figure 1A: Use colours to highlight sequence identities and similarities, improve resolution.

Experimental design

Figure 2: The co-localisation of nuclear marker and P2 seems inhomogeneous, especially in the protoplast. I suggest including a normalised intensity profile for nuclear localization in panels A and B.

Figure 4A: Indicate input & digested reactions in the panels. If the left lane for each sample is the undigested input and the right lane is digested, both input and digested sample look similar. Include a positive control to verify the endonuclease activity of both enzymes used in the assay.

Figure 4B/C: The last sentence states that each experiment has been repeated at least three times. Please state if the same amount of data/measurements was included in the statistics for all samples.

Figure S1: The upper (a-c) and middle (d-f) rows show the plants from different perspectives. I suggest showing side-by-side images or use the same perspective for all plants/samples.

Methods:
Lines 154-155: Were both endonucleases used in the same reaction or in individual reactions? Please state this clearly.
Lines 156-158: For how long were the digestion reactions incubated?

Validity of the findings

Discussion:
Lines 339-340: Has it been shown elsewhere/is there evidence that the insertion in BSV-OL-P2 is truly part of the coiled coil domain? In addition, if there is low sequence similarity/conservation among the P2 representatives in this study, it seems inappropriate to base the prediction of the localisation of the coiled-coil domain on a sequence alignment. It could be useful to run InterPro for all representatives in the alignment individually to see if the results coincide with the sequence alignment.

Conclusion: The statement about the influence of P2 on DNA methylation needs to be supported by clearer experimental evidence.

---

## Round 0.2 · Minor Revisions

I have read your revision and the rebuttal letter. I think that you have addressed the reviewers comments appropriately in your rebuttal letter. However, you have not always incorporated this information into your manuscript. Thus, the revised manuscript has not yet sufficiently improved for publication.

Please incorporate the following information of your responses in the rebuttal letter into your manuscript:

Reviewer 1 / Response 2 (yield loss), Response 6 (number of Isolates),
Reviewer 2 / Response 3 (introduce example viruses), Response 4 (connection between RTBV and Badnaviruses), Response 10 (same amount of data), Response 13 (conditions of digestion reaction)

You have not adequately addressed
Reviewer 2 Point 7 (color in figure 1): The request is not about the square around the coiled coils but about the annotation of the alignment. The inverted black squares are not readable. You might want to use the clustal color scheme or something equivalent. Please revise
Reviewer 2 Point 15 (conclusion of DNA Methylation): If you will show evidence for the role of P2 in DNA-Methylation in future, and do not want to show/cite these results now, than you should not draw the conclusion on DNA-methylation in this manuscript here. Please rephrase (either remove or cite additional evidence)
.
For the points listed above, please change your manuscript accordingly and incorporate your comments from the rebuttal letter into the manuscript. These should be minor edits that can be quickly addressed in a final revision.

---

## Round 0.3 · accepted · Accept

I have carefully read your re-revision. You you have addressed the reviewers concerns and included the missing information in the manuscript, The manuscript is now ready for publication.